# Robust Expected Information Gain for Optimal Bayesian Experimental Design Using Ambiguity Sets

**Jinwoo Go**[1]

**Tobin Isaac**[1]

[1]Computational Science and Engineering Dept., Georgia Institute of Technology, Atlanta, Georgia, USA

## Abstract

The ranking of experiments by expected information gain (EIG) in Bayesian experimental design is sensitive to changes in the model's prior distribution, and the approximation of EIG yielded by sampling will have errors similar to the use of a perturbed prior. We define and analyze *robust expected information gain* (REIG), a modification of the objective in EIG maximization by minimizing an affine relaxation of EIG over an ambiguity set of distributions that are close to the original prior in KL-divergence. We show that, when combined with a sampling-based approach to estimating EIG, REIG corresponds to a 'log-sum-exp' stabilization of the samples used to estimate EIG, meaning that it can be efficiently implemented in practice. Numerical tests combining REIG with variational nested Monte Carlo (VNMC), adaptive contrastive estimation (ACE) and mutual information neural estimation (MINE) suggest that in practice REIG also compensates for the variability of under-sampled estimators.

## 1 INTRODUCTION

Bayesian Experimental Design (BED) is a probabilistic framework for selecting experiments to learn about one or more uncertain variables. Within BED, the most popular criterion for ranking experiments is by Expected Information Gain (EIG), which estimates from current knowledge, encoded in a prior distribution, how informative a particular experiment is likely to be. This framework is used in diverse applications across many disciplines [Ryan et al., 2016a], in natural sciences [Huan, 2010], social sciences [Embretson and Reise, 2013], and in machine learning and data analysis [Foster et al., 2020].

The sensitivity of experimental design to misspecification of the prior distribution has been described in [DasGupta and Studden, 1991, Ryan et al., 2016b], even in settings where the information gain is computable in closed form. For more complex models, EIG can only be estimated numerically, which may further affect the reliability of the computed rankings of experiments (or, in the case of continuously parameterized experiments, the gradient of EIG). EIG is by definition an expectation of an expectation, so general purpose estimates, such as Nested Monte Carlo (NMC) estimation [Ryan, 2003], can be expensive, slow to converge, and sensitive to underconverged sample estimates.

To address the issues above that affect the reliability of EIG estimates in BED, we introduce a quantity we call *robust expected information gain* (REIG) as a probability-theoretic way of ranking experiments by their expected information gain for some worst-case small perturbation of the prior. We also show through convex analysis that the estimation of REIG is a simple post-processing of the samples generated by an NMC-like estimator. As a result, our methodology is applicable with many existing methods, which we demonstrate in section 7 by applying REIG to samples generated by three recent popular EIG estimators [Foster et al., 2020, 2019, Kleinegesse and Gutmann, 2020].

## 2 BACKGROUND AND NOTATION

We use $\theta \in \Theta$ to indicate a choice of parameters for a model from a set of possible parameters, and we assume a reference prior probability distribution of $\theta$ which has a measurable density function $p(\theta)$, so that we can write $E_{p(\theta)}[f] = \int_\Theta f(\theta)p(\theta) \, d\theta$. We let $\xi \in \Xi$ be a potential experiment from a class of experiments, which has an outcome variable $y(\xi)$. The experiment $\xi$ is modeled by the likelihood function $p(y|\theta, \xi)$, which for each choice of $\theta$ defines a measurable probability density function of $y$. Our interest is in models where the densities $p(\theta)$ and $p(y|\theta, \xi)$ can be efficiently computed, and where samples can be drawn from $p(\theta)$ and from $p(y|\theta, \xi)$ for each $(\theta, \xi)$, so that the joint prior distribution $p(\theta, y|\xi) = p(\theta)p(y|\theta, \xi)$ also

*Accepted for the 38th Conference on Uncertainty in Artificial Intelligence* (UAI 2022).

has a computable density function and can be sampled. We use the notation $p(y|\xi) = E_{p(\theta)}[p(y|\theta,\xi)]$ for the marginal distribution of the outcome $y$.

## 2.1 PRIOR UNCERTAINTY

While it seems recursive to consider uncertainty in the prior distribution used in Bayesian inference, the prior distribution is in many settings not determined from first principles or an existing population of data. In these cases the choice of prior is often dictated by what is required to make a computation tractable or simple, by invariance principles, or by an attempt to be noninformative [Stark, 2015]. But notions of noninformative priors do not scale to high dimensions [Yang and Berger], and even in low dimensions priors that are close under a weak topology like total variation can have diverging posteriors for the same observations [Owhadi et al., 2015].

So in this work we will consider sets of prior distributions $q(\theta)$ other than the reference $p(\theta)$, but we only consider $q(\theta)$ that are absolutely continuous with respect to $p(\theta)$. Although methods similar to ours are used to handle model uncertainty [Shapiro et al., 2021], we treat $p(y|\theta,\xi)$ as certain: the only uncertainty we consider is in the prior distribution of $\theta$. When we extend the notation of derived distributions from $p(\theta)$ to another $q(\theta)$, then, it is always with the same likelihood: the joint prior $q(\theta, y|\xi) = q(\theta)p(y|\theta,\xi)$, the marginal $q(y|\xi) = E_{q(\theta)}[p(y|\theta,\xi)]$, etc.

## 2.2 EXPECTED INFORMATION GAIN

We use $D_{\mathrm{KL}}(p(\theta)\|q(\theta))$ to denote the Kullback-Leibler divergence, $D_{\mathrm{KL}}(p(\theta)\|q(\theta)) = \int_\Theta p(\theta)\log(p(\theta)/q(\theta))\,d\theta$. The expected information gain of experiment $\xi$ is defined to be the expectation over the marginal distribution of outcomes $p(y|\xi)$ of the KL-divergence from the Bayesian posterior distribution $p(\theta|y,\xi)$ to the prior $p(\theta)$. Because the prior is not fixed in this work, we consider EIG to be a function of both the prior $p(\theta)$ and the experiment $\xi$,

$$I(p,\xi) = E_{p(y|\xi)}[D_{\mathrm{KL}}(p(\theta|y,\xi)\|p(\theta))]. \quad (1)$$

Bayesian optimal experimental design seeks the experiment $\xi^*$ that maximizes this quantity,

$$\xi^* = \arg\max_{\xi \in \Xi} I(p,\xi).$$

Although the form of EIG in (1) is the most intuitive for Bayesian experimental design, other equivalent definitions map more directly on the robust variant we introduce in section 5 and the sampling-based estimators in section 6. The EIG of an experiment is also the mutual information

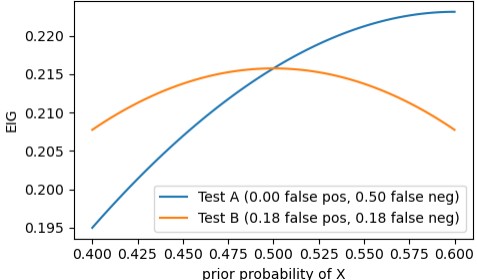

Figure 1: The expected information gain of two tests for condition X depends on the condition's prior probability.

between $\theta$ and $y$,

$$I(p,\xi) = D_{\mathrm{KL}}(p(\theta,y|\xi)\|p(\theta)p(y|\xi)) \quad (2)$$
$$= E_{p(\theta)}[D_{\mathrm{KL}}(p(y|\theta,\xi)\|p(y|\xi))] \quad (3)$$
$$= E_{p(\theta,y|\xi)}\left[\log\frac{p(y|\theta,\xi)}{p(y|\xi)}\right]. \quad (4)$$

The last form in (4) is the preferred form for many methods that estimate EIG when the likelihood $p(y|\theta,\xi)$ and prior $p(\theta)$ can be evaluated directly. We discuss methods for estimating $I(p,\xi)$ and other related quantities in section 6.

## 3 A SIMPLE EXAMPLE WITH TWO EXPERIMENTS

Suppose a doctor has two blood tests for Condition X: test A has a $10^{-14}\%$ chance of a false negative but a $50\%$ chance of a false positive, and test B has an $\approx 18.4\%$ chance of a false negative and the same chance of a false positive. If the doctor estimates the prior probability that a patient has condition X is $50\%$, it turns out that both tests have the same expected information gain of $\approx 0.22$ nats. If that prior probability could be mistaken, however, the two tests have different EIGs for prior probabilities in the vicinity of $50\%$, shown in fig. 1. If the prior probability the patient has Condition X is actually $> 50\%$, then test A has a greater EIG than test B, and vice versa if it is $< 50\%$.

We interpret these results as follows: For test A, if the prior probability of Condition X is $> 50\%$, then a negative test result is surprising because there are essentially no false negatives, but if the prior probability is $< 50\%$, then a positive result is less surprising because it has a high probability of being a false positive. So in comparison to test B, the EIG of test A is more sensitive to the choice of prior. In any neighborhood of $p = 50\%$, there are priors where test A is expected to be less informative than test B. So one could argue that a risk-averse doctor, who would maximize how informative the test would be in the worst case, should select test B.

## 4 AMBIGUITY SETS

In the simple example above, we used a range of prior probabilities for the model parameter to argue that some tests are less locally sensitive to perturbations of the prior distribution. To generalize this idea from a simple discrete example to other probability distributions, we rely on the notion of an *ambiguity set* [Bayraksan and Love, 2015, Watson and Holmes, 2016], which is a set of distributions that are not far from a reference prior $p(\theta)$ in some statistical distance. We use KL-divergence as distance, so our ambiguity set with radius $\epsilon$ centered at reference distribution $p(\theta)$ is

$$\mathcal{A}(\epsilon, p) = \{q : D_{\mathrm{KL}}(q(\theta) \| p(\theta)) \le \epsilon\}.$$

KL-divergence as a distance works well with Bayesian optimal experimental design, because the KL-divergence appears in the definition of EIG, and because the set $\mathcal{A}(\epsilon, p)$ is defined as a convex subset of positive measurable functions $q(\theta)$ with just two conditions: $\int_\Theta q(\theta)\, d\theta = 1$ and $D_{\mathrm{KL}}(q(\theta) \| p(\theta)) \le \epsilon$. Thus the minimization over $q \in \mathcal{A}(\epsilon, p)$ of a well-behaved convex function $f(q)$, which appears difficult because $\mathcal{A}(\epsilon, p)$ of the infinite dimensionality of space of measurable functions, transforms into an equivalent dual convex program with only two variables.

We direct the interested reader to [Shapiro, 2017] for additional details: here we summarize the results that are important for this work. If the objective function of interest $f(q)$ is the expectation under $q(\theta)$ of a measurable quantity of interest $Z(\theta)$, then it is an affine function of $q$ and we may use duality to simplify the maximization or minimization of $E_{q(\theta)}[Z(\theta)]$ over $\mathcal{A}(\epsilon, p(\theta))$ into a dual problem with only one variable $\lambda \ge 0$. The maximization problem

$$R_\epsilon = \sup_{q \in \mathcal{A}(\epsilon, p)} [Z(\theta)] \tag{5}$$

can be solved in dual form as

$$R_\epsilon = \inf_{\lambda \ge 0} \lambda\epsilon + \lambda \log E_{p(\theta)}[\exp(\lambda^{-1} Z(\theta))], \tag{6}$$

and the minimization problem

$$M_\epsilon = \inf_{q \in \mathcal{A}(\epsilon, p)} [Z(\theta)] \tag{7}$$

can be solved in dual form as

$$M_\epsilon = -\inf_{\lambda \ge 0} \lambda\epsilon + \lambda \log E_{p(\theta)}[\exp(-\lambda^{-1} Z(\theta))]. \tag{8}$$

It is important to note that these are non-parametric results. Given a parameterized family of priors $\{p(\theta; \psi)\}_\psi$, the gradient $\nabla_\psi E_{p(\theta; \psi)}[Z(\theta)]$ is sufficient to compute the optimizer in $\mathcal{A}(\epsilon, p)$ of (5) or (7) within the parametric family because the objective is affine. But (6) and (8) allow us to compute the optimal objective value over the entire ambiguity set without explicitly computing an optimal distribution in the set.

## 5 ROBUST BAYESIAN EXPERIMENTAL DESIGN

The insight of the example of section 3 was that a risk-averse approach to experimental design that allows for some uncertainty in the prior distribution would select the experiment that maximizes the worst-case EIG in the vicinity of the reference prior $p(\theta)$. Using the ambiguity set $\mathcal{A}(\epsilon, p(\theta))$ from section 4 to define the vicinity of $p(\theta)$, we first formalize the worst-case EIG as $I_\epsilon^{\mathrm{true}}(p, \xi)$, the *true robust expected information gain with radius $\epsilon$*,

$$I_\epsilon^{\mathrm{true}}(p, \xi) = \inf_{q \in \mathcal{A}(\epsilon, p)} I(q, \xi). \tag{9}$$

The experiment that maximizes this quantity is

$$\begin{aligned} \xi_{\mathrm{REIG}, \epsilon, \mathrm{true}}^* &= \arg\max_{\xi \in \Xi} I_\epsilon^{\mathrm{true}}(p, \xi) \\ &= \arg\max_{\xi \in \Xi} \inf_{q \in \mathcal{A}(\epsilon, p)} I(q, \xi). \end{aligned}$$

This optimization problem in this definition has a clear meaning, but we note that $I(q, \xi)$ is a quantity that is concave in $q$, so it can have multiple local minima in the convex ambiguity set $\mathcal{A}(\epsilon, p)$ and the duality framework of section 4 cannot be applied directly.

### 5.1 AFFINE EXPECTED INFORMATION GAIN APPROXIMATION

To define a relaxation to a tractable problem, we split $I(q, \xi)$ into two contributions, one with the marginal distribution of $y$ fixed by the reference prior, $y \sim p(y|\xi)$, and the other a correction that is the divergence between $p(y|\xi)$ and $q(y|\xi)$,

$$\begin{aligned} I(q, \xi) &= E_{q(\theta, y|\xi)} \big[ \log \big( \frac{p(y|\theta, \xi)}{p(y|\xi)} \frac{p(y|\xi)}{q(y|\xi)} \big) \big] \\ &= E_{q(\theta, y|\xi)} \big[ \log \frac{p(y|\theta, \xi)}{p(y|\xi)} \big] \\ &\quad - D_{\mathrm{KL}}(q(y|\xi) \| p(y|\xi)). \end{aligned} \tag{10}$$

We denote the first term in this difference

$$I_{\mathrm{aff}}(q, \xi; p) = E_{q(\theta, y|\xi)} \big[ \log \frac{p(y|\theta, \xi)}{p(y|\xi)} \big], \tag{11}$$

because it is an approximation to $I(q, \xi)$ that is affine and exact when $q = p$. By the concavity of EIG with respect to its first argument q,

$$I_{\mathrm{aff}}(q, \xi; p) \ge I(q, \xi) \quad \text{for all } q. \tag{12}$$

Due to the data processing inequality that the mutual information between two random variables cannot increase by a deterministic or random transformation of the arguments, the error in $I(q, \xi; p)$ is bounded by

$$\begin{aligned} |I(q, \xi) - I_{\mathrm{aff}}(q, \xi; p)| &\le D_{\mathrm{KL}}(q(y|\xi) \| p(y|\xi)) \\ &\le D_{\mathrm{KL}}(q(\theta) \| p(\theta)). \end{aligned} \tag{13}$$

In fact $I_{\text{aff}}(q, \xi; p)$ is the affine approximation to $I(q, \xi)$ that is tangent at $q = p$.

**Theorem 1.** *The function $I_{\text{aff}}(q, \xi; p)$ from (11) is tangent to $I(q, \xi)$ at $q = p$ for every design $\xi$.*

*Proof.* It is sufficient to show that the difference between the two functions, which by (10) is $D_{\text{KL}}(q(y|\xi)\|p(y|\xi))$, is gradient free at $q = p$.

We first calculate the gradient with respect to $\xi$: by the chain rule applied to (4), the derivative in the direction $\hat{\xi}$ is

$$\nabla_\xi D_{\text{KL}}(q(y|\xi)\|p(y|\xi))[\hat{\xi}]$$
$$= \nabla_{q(y|\xi)} D_{\text{KL}}(q(y|\xi)\|p(y|\xi))[\nabla_\xi q(y|\xi)[\hat{\xi}]]$$
$$- E_{q(y|\xi)}\Big[\frac{\nabla_\xi p(y|\xi)[\hat{\xi}]}{p(y|\xi)}\Big].$$

For general distributions $Q$ and $P$ the KL divergence satisfies $\nabla|_{Q=P} D_{\text{KL}}(Q\|P) = 0$, so the first term vanishes when $q = p$. In the second term, when $q = p$ the denominator cancels with the measure and we have

$$E_{q(y|\xi)}\Big[\frac{\nabla_\xi p(y|\xi)[\hat{\xi}]}{p(y|\xi)}\Big]\Big|_{q=p} = \int \nabla_\xi p(y|\xi)[\hat{\xi}] \, dy$$
$$= \nabla_\xi (E_{p(y|\xi)}[1])[\hat{\xi}] = 0,$$

where we use the fact that $E_{p(y|\xi)}[1] = 1$ for all $\xi$.

Finally, we can see that the gradient with respect to $q(\theta)$ in the direction $\hat{q}(\theta)$ is

$$\nabla_{q(\theta)} D_{\text{KL}}(q(y|\xi)\|p(y|\xi))[\hat{q}(\theta)]$$
$$= \nabla_{q(y|\xi)} D_{\text{KL}}(q(y|\xi)\|p(y|\xi))[\nabla_{q(\theta)} q(y|\xi)[\hat{q}(\theta)]],$$

which vanishes at $q = p$ for the same reasons as above. $\qquad\square$

## 5.2 ROBUST EXPECTED INFORMATION GAIN (REIG)

Having shown that $I_{\text{aff}}(q, \xi; p)$ is a good approximation to $I(q, \xi)$ near the reference prior $p(\theta)$, we now use it to define a robust quantity that approximates $I_\epsilon^{\text{true}}$, which we refer to simply as $I_\epsilon$,

$$I_\epsilon(p, \xi) = \inf_{q \in \mathcal{A}(\epsilon, p)} I_{\text{aff}}(q, \xi; p). \qquad (14)$$

By the properties established in (12), (13), and theorem 1, we have the following relationships between $I_\epsilon^{\text{true}}$ and $I_\epsilon$:

$$I(p, \xi) \geq I_\epsilon(p, \xi) \geq I_\epsilon^{\text{true}}(p, \xi); \qquad (15)$$

$$|I_\epsilon(p, \xi) - I_\epsilon^{\text{true}}(p, \xi)| \leq \epsilon; \qquad (16)$$

$$|I_\epsilon(p, \xi) - I_\epsilon^{\text{true}}(p, \xi)| \in O(\epsilon^2). \qquad (17)$$

These facts suggest an experiment that maximizes $I_\epsilon(p, \xi)$,

$$\xi_{\text{REIG}, \epsilon}^* = \arg\max_{\xi \in \Xi} I_\epsilon(p, \xi),$$

has similar robustness to $\xi_{\text{REIG}, \epsilon, \text{true}}^*$ over perturbations of the the prior $p(\theta)$, as long as the radius $\epsilon$ of the ambiguity set is not too large.

## 5.3 COMPUTATION OF $I_\epsilon(p, \xi)$ VIA DUALITY

We have selected $I_\epsilon(p, \xi)$ as our robust quantity to optimize because (14) can be optimized by the dual transformation described in section 4. Applying (8) to EIG in the form (3), we have $I_\epsilon(p, \xi) =$

$$- \inf_{\lambda \geq 0} \lambda \epsilon + \lambda \log E_{p(\theta)}\Big[ \exp\Big( -\frac{D_{\text{KL}}(p(y|\theta, \xi)\|p(y|\xi))}{\lambda} \Big)\Big]. \qquad (18)$$

We will show in section 6 that this 1D convex optimization problem can be solved efficiently by a small adaptation of existing EIG estimators.

## 5.4 RELATED DESIGN CRITERIA

Our definition of $I_\epsilon$ was motivated by a risk-aversion argument in favor of the design with the best worst-case EIG in a neighborhood. Because the approximation $I_{\text{aff}}(q, \xi; p)$ is affine, however, maximization over the ambiguity set can also be solved by duality. This means that the same methodology can be used to define a risk-loving strategy for experimental design, which selects the experiment that has the highest EIG for some prior in the ambiguity set. We call this criterion $I_{\epsilon, \max}(p, \xi) =$

$$\inf_{\lambda \geq 0} \lambda \epsilon + \lambda \log E_{p(\theta)}\Big[ \exp\Big( \frac{D_{\text{KL}}(p(y|\theta, \xi)\|p(y|\xi))}{\lambda} \Big)\Big]. \qquad (19)$$

This criterion is used in section 7 to counteract biased underestimation of EIG by some estimators.

Last, we note that our decision to limit the uncertainty in the models of the experiments to just the prior $p(\theta)$ and not the likelihood $p(y|\theta, \xi)$ is arbitrary, at least from the perspective of the methods we have developed. An ambiguity set $\mathcal{A}(\epsilon, p(\theta, y|\xi))$ can be centered around the joint prior of the model $p(\theta, y|\xi)$, and an affine approximation can be taken that would allow for optimization over that ambiguity set via duality. The result would be an even more conservative quantity, $I_{\epsilon, \text{joint}}(p, \xi) =$

$$- \inf_{\lambda \geq 0} \lambda \epsilon + \lambda \log E_{p(\theta, y|\xi)}\Big[ \exp\Big( \lambda^{-1} \log \frac{p(y|\xi)}{p(y|\theta, \xi)} \Big)\Big]. \qquad (20)$$

We will not explore this criterion more in this work.

## 6 REIG ESTIMATION VIA SAMPLING

The design of efficient estimators for EIG has been the subject of much research, in part because their use in experimental design is computationally demanding. The combination of nested iterations to estimate the densities of implicitly defined distributions, to evaluate the expectation of the EIG, and finally to optimize that quantity lead to many passes over the problem data as well as many model evaluations.

When introducing an implicitly defined quantity like $I_\epsilon$, we should be leery of adding another nested loop to the calculation. This is why we immediately discounted the design criterion $I_\epsilon^{\text{true}}$ from (9), which would require optimization in the original variables parameterizing $p(\theta)$, which could be numerous.

## 6.1 CONSTRUCTING A REIG ESTIMATOR

When both the prior $p(\theta)$ and the likelihood $p(y|\theta, \xi)$ can be sampled directly, sampling-based approaches to estimating EIG often have a two-level structure: an inner estimator is defined for a fixed $\theta$ and/or $y$ in the integrated quantity — either $D_{\text{KL}}(p(\theta|y, \xi)\|p(\theta))$ in (1), $D_{\text{KL}}(p(y|\theta, \xi)\|p(y|\xi))$ in (3), or $\log(p(y|\theta, \xi)/p(y|\xi))$ in (4) — and an outer Monte Carlo estimator over either $p(\theta)$ or $p(\theta, y|\xi)$ calls the inner estimator for each generated $\theta$ or $(\theta, y)$.

This basic paradigm maps closely onto the dual formulation of $I_\epsilon$ in (18), in a method we sketch in algorithm 1 that defines a REIG estimator $\hat{I}_\epsilon$.

1. Draw $N_1$ i.i.d. samples $\{\theta_i\}_{i=1}^{N_1}$ from $p(\theta)$.
2. For each $\theta_i$, use estimator $\tilde{D}(\theta, \xi)$ to compute an estimate $d_i \leftarrow \tilde{D}(\theta_i, \xi)$ of $D_{\text{KL}}(p(y|\theta, \xi)\|p(y|\xi))$.
3. Solve the 1D convex optimization problem

$$M_\epsilon = \inf_{\lambda \geq 0} \lambda\epsilon + \lambda \log \frac{1}{N_1} \sum_{i=1}^{N_1} \exp(-\lambda^{-1} d_i) \quad (21)$$

and return $-M_\epsilon$.

**Algorithm 1:** $I_\epsilon$ Estimation via Sampling

In this approach the inner estimator is called $N_1$ times in step 2, which is the same number of times it would have been called to compute the EIG estimator $\frac{1}{N_1} \sum_{i=1}^{N_1} d_i$, but those estimates are saved and treated as an empirical distribution, so that the optimization problem in step 3 solves (18) by sample average approximation (SAA) instead of stochastic approximation (SA). The assumption is that this 1D convex problem will be solved quickly and the dominant cost in algorithm 1 is the cost of computing the KL-divergence estimators $d_i \leftarrow \tilde{D}(\theta_i, \xi)$.

Although the inner optimization in step 3 is solved by SAA, we note that algorithm 1 can be used within either SAA or SA for the optimization over $\xi$, depending on whether the samples in step 1 are reused or not. The derivative $\nabla_\xi \hat{I}_\epsilon(p, \xi)$ can be computed as $-\nabla_d M_\epsilon \cdot \nabla_\xi d$, where $d$ is the vector of $d_i$ estimates from step 2. The partial derivatives $\nabla_\xi d$ are also present in computing the gradients of EIG estimators, so existing methods for this term can be reused: $\nabla_d M_\epsilon$ are the only additional derivatives needed for REIG. Letting $\lambda^*$ be the optimal value and letting $L(d) = \log \frac{1}{N_1} \sum_{i=1}^{N_1} \exp(-d_i)$, if $\lambda^* \neq 0$ then $\nabla_d M_\epsilon =$

$\nabla_d L(d/\lambda^*)$. If $\lambda^i = 0$, there is some $i^* = \arg\min_i d_i$ and $M_\epsilon = -d_i$, so that either $-e_i = \nabla_d M_\epsilon$ or $-e_i \in \partial_d M_\epsilon$ if $i^*$ is not unique.

## 6.2 EIG ESTIMATORS

There are many possible choices for the estimator $\tilde{D}(\theta, \xi)$ of $D_{\text{KL}}(p(y|\theta, \xi)\|p(y|\xi))$. In fact, any EIG estimator $\hat{D}(p(\theta), p(y|\theta, \xi))$ that accepts general prior distributions can be used by running a separate instance of $\hat{D}$ for each $\theta_i$ with the prior distribution $\theta \sim \delta_{\theta_i}$. In practice, this approach would have poor performance because sequestering the samples $\theta_i$ into separate estimator instances would not allow for vectorization across samples. Vectorization and batching are best exploited in a nested EIG estimator if all instances of the inner estimator have the same hyperparameters to maximize throughput.

In the experiments in section 7, we have primarily used the estimators developed in [Foster et al., 2019, 2020, Kleinegesse and Gutmann, 2020]. The first is the *Variational Nested Monte Carlo* (VNMC) estimator, which is based on the form of EIG in (4). It draws $N$ samples $(\theta_i, y_i) \sim p(\theta, y|\xi)$ from the joint prior, evaluates $\log p(y_i|\theta_i, \xi)$ directly, and then uses an inner estimator for $\log p(y_i|\xi)$. That estimator is based on the identity

$$\log p(y|\xi) = \log E_{q(\theta)}\left[\frac{p(\theta)}{q(\theta|y, \xi)} p(y|\theta, \xi)\right],$$

where $q(\theta|y, \xi)$ can be any distribution that is absolutely continuous with respect to $p(\theta)$, but the variance of that expectation is lower the closer $q(\theta|y, \xi)$ is to the posterior distribution $p(\theta|y, \xi)$. $M$ samples $\{\theta_i^j\}_{j=1}^M$ are drawn from $q(\theta|y, \xi)$ for each $y_i$, resulting in the EIG estimator

$$\hat{I}^{\text{VNMC}} = \sum_{i=1}^N \log \frac{p(y_i|\theta_i, \xi)}{\frac{1}{M} \sum_{j=1}^M \frac{p(\theta_i^j)}{q(\theta_i^j|y_i, \xi)} p(y_i|\theta_i^j, \xi)}. \quad (22)$$

This estimator is consistent in the limit as $M \to \infty$, but for finite $M$ is in expectation an upper bound for EIG. For details see Foster et al. [2019].

The second estimator that we use is the *Adaptive Contrastive Estimator* (ACE) from [Foster et al., 2020]. In description it is almost identical to VNMC, except that in estimating $p(y_i|\xi)$ we add to the samples $\{\theta_i^j\}_{j=1}^M$ the original sample $\theta_i^0 = \theta_i$ drawn from $p(\theta)$ that generated $y_i$. The result is

$$\hat{I}^{\text{ACE}} = \sum_{i=1}^N \log \frac{p(y_i|\theta_i, \xi)}{\frac{1}{M+1} \sum_{j=0}^M \frac{p(\theta_i^j)}{q(\theta_i^j|y_i, \xi)} p(y_i|\theta_i^j, \xi)}. \quad (23)$$

This is also a consistent estimator of EIG, but the addition of the prior-samples $\theta_i^0$ to the estimator for $p(y_i|\xi)$ makes it in expectation a lower bound for EIG for finite $M$: see [Foster et al., 2020] for more details.

The last estimator that we use is the *Mutual Information Neural Estimation* (MINE), which trains the ratio, $\frac{p(y|\theta,\xi)}{p(y|\xi)}$, with samples and estimate the EIG with SAA method [Kleinegesse and Gutmann, 2020],

$$\hat{I}^{\text{MINE}} = \sum_i^N [T_\psi(\theta_i, y_i) - e^{T_\psi(\theta_i, y_i^*)-1}], \quad (24)$$

where $y_i^*$ represents the shuffled $y$ and $T$ is a neural network with parameters $\psi$.

The outer loop for ACE, VNMC and MINE methods draw from $p(\theta, y|\xi)$ or $p(\theta)p(y|\xi)$, and in the implementations provided by the authors one sample from $p(y|\theta_i, \xi)$ is drawn for each of $N$ samples $\theta_i$ drawn from $p(\theta)$. To adapt these methods to the needs of our algorithm 1, we split $N$ into $N = N_1 N_2$, and draw $N_2$ samples from $p(y|\theta_i, \xi)$ for each of $N_1$ samples drawn from $p(\theta)$. We then take the mean over the $N_2$ samples for our estimate of $D_{\text{KL}}(p(y|\theta_i, \xi)\|p(y|\xi))$ on line 2 of our algorithm.

### 6.3  REIG AS LOG-SUM-EXP STABILIZATION

Step 3 of algorithm 1 shows that the convex dual objective function for the $I_\epsilon$ design criterion manifests as a $\lambda\epsilon$-biased and $\lambda^{-1}$-weighted log-sum-exp combination of the $-D_{\text{KL}}(p(y|\theta,\xi)\|p(y|\xi))$ estimates.

From the bounds for log-sum-exp operators we have

$$\lambda\epsilon - \min_i\{d_i\} \geq \lambda\epsilon + \lambda \log \frac{1}{N} \sum_{i=1}^N \exp(-\lambda^{-1}d_i)$$
$$> \lambda\epsilon - \min_i\{d_i\} - \lambda \log N.$$

When the ambiguity set radius $\epsilon$ is smaller, the optimal $\lambda$ is larger, and $M_\epsilon$ becomes more like the sample mean; when $\epsilon$ is larger, the optimal $\lambda$ is smaller until eventually the $\lambda \geq 0$ constraint becomes binding. In that case the limit as $\lambda \to 0$ is achieved and the value is squeezed to become $M_\epsilon = -\min_i \ d_i$.

We interpret these facts in the following way: $I_\epsilon$ tends to bias the samples in the EIG estimate more towards the smaller values of $D_{\text{KL}}(p(y|\theta_i, \xi)\|p(y|\xi))$ the larger $\epsilon$ is. A well-recognized problem in EIG estimation [Foster et al., 2020] is the presence of samples where $p(y|\xi)$ is under-estimated, leading to artificially inflated EIG estimates.

We have until this point consider $I_\epsilon$ a design criterion in its own right, but this biasing behavior suggests that the use of algorithm 1 with an appropriate choice of $\epsilon$ can also be useful as an estimator for the original EIG criterion whose bias protects the estimate from sampling error. This may be a viable approach for stabilizing the computed EIG value when there are insufficient samples to converge the estimate of $p(y|\xi)$ well in the nested sampling approaches.

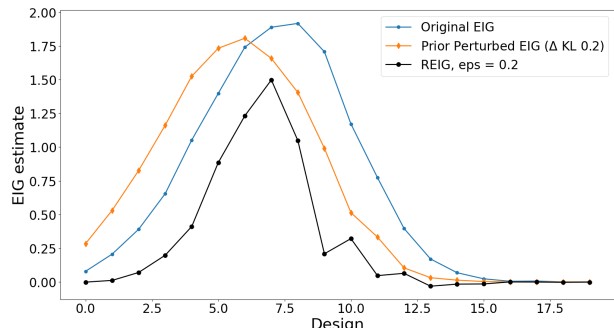

Figure 2: Preference test: EIG change with prior perturbation and the performance of $I_\epsilon$

Since $\hat{I}^{\text{VNMC}}$ is an upper bound while $\hat{I}^{\text{ACE}}$ and $\hat{I}^{\text{MINE}}$ are lower bounds for $I$, we define an estimator $\hat{I}_\epsilon^{\text{VNMC}}$ that uses VNMC as the estimator for (18) and the estimators $\hat{I}_{\epsilon,\text{max}}^{\text{ACE}}$ and $\hat{I}_{\epsilon,\text{max}}^{\text{MINE}}$ that use ACE and MINE, respectively. The minimization / maximization in each case acts counter to the bias of the sampling process, which we measure in the following section.

## 7  EXPERIMENTS

We perform two types of numerical experiments. In the first type, we test to see how well $I_\epsilon$ performs as a "worst-case" EIG estimate, as discussed in section 5. In the second type, we compare the previously developed EIG estimators from section 6.2 with a large number of samples to $I_\epsilon$ with a small number of samples to see if the bias against large summands described in section 6.3 is as effective as additional samples in stabilizing the computation.

### 7.1  REIG AS A WORST-CASE ESTIMATE

This test uses the Preference model test case from [Foster et al., 2019]. The parameter $\theta$ is a location parameter with a reference prior $p(\theta)$ that is normally distributed, and the experiments are indexed by locations relative to $\theta$.

To test how well $I_\epsilon$ serves at computing worst-case estimate, we select as second prior $q(\theta)$ such that $I(p)$ and $I(q)$ look visibly distinct over the range of possible experiments in fig. 2, and then measure the KL-divergence between $p(\theta)$ and $q(\theta)$ and take this number to be $\epsilon$ (in this case, $\epsilon = 0.2$). We then compute the $I_\epsilon$ criterion for all experiments as well, and plot them in fig. 2. (In all of these calculations we use a sampling estimator with a large number of samples so we can be reasonably sure that the values are converged.) What we see is that $I_\epsilon(p, \xi)$ succeeds at being a lower bound for both $I(p, \xi)$ and $I(q, \xi)$ in this case. We also see that this is not the result of a uniform or log-uniform scale reduction:

$I_\epsilon(p,\xi)$ has more drastically reduced the gain of some experiments than others, meaning that $I_\epsilon(p,\xi)$ determines a different optimal $\xi^*$ than $I(p,\xi)$.

## 7.2 EIG STABILIZATION VIA REIG

We now test the effectiveness of $\hat{I}_\epsilon(p,\xi)$'s log-sum-exp stabilization at producing a stabilized estimation of $I(p,\xi)$. We use the benchmarks designed by [Foster et al., 2019, Kleinegesse and Gutmann, 2020], especially three experimental designs which have explicit models for the likelihood distribution: A/B test, Preference, and Pharmacokinetic model.

### 7.2.1 A/B test

An A/B test [Kohavi et al., 2009, Box et al., 1978] can be used to determine which experiment in a set of two or more results in the largest information gain. Foster et al. [2019] introduces the group size selection problem between groups A and B. We have $n$ experiment participants, and we can select $n_A$ participants for group A and $n - n_A$ participants for group B. Each participant is represented as two random variables: the first is measured for participants in group A, and the second for group B. This A/B test models each participant's random variables using a Bayesian linear model, $y = X\theta + \epsilon$; the prior and likelihood distributions are Gaussian distributions.

Each of the estimators under consideration uses a neural network to generate samples that musts be trained. The proposal distribution used by the VNMC and ACE estimators is trained with $A$ and $\Sigma_p$ as in [Foster et al., 2019]: more details about the proposal distribution can be found there. Likewise, the parameters $\psi$ of the neural network in $\hat{I}^{\text{MINE}}$ are trained for each $\theta$ with samples of $y$ from the given distributions.

In our work, we start from a different reference prior distribution, with its mean taken to be $[4.46, 0]$ instead of $[0, 0]$. We find that the VNMC estimator shows considerably more error for some designs with a small number of samples $M = 30$ (fig. 3, top left), and requires more samples to converge ($M \geq 100$). MINE, on the other hand, doesn't have big difference when we have enough samples (fig. 3, bottom left).

In contrast to that larger number of samples, we compute $\hat{I}_\epsilon^{\text{VNMC}}$ and $\hat{I}_{\epsilon,\max}^{\text{ACE}}$ estimators using only $M = 30$ posterior samples and $\hat{I}_{\epsilon,\max}^{\text{MINE}}$ using $1000*10$ samples, with an increasing series of ambiguity set radii $\epsilon$ (fig. 3, right). The estimate is almost unaffected by the ambiguity set if $\epsilon = 0.001$, but increasing $\epsilon$ to 0.1 seems to improve the estimates for the designs that were highly overestimated without negatively affecting the other experiments that were already properly estimated.

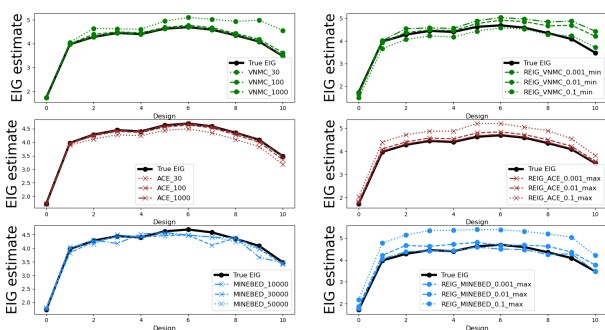

Figure 3: A/B test convergence for EIG estimators (left column): 100*10 samples from $p(\theta, y)$ and 30/100/1000 samples from $q_\phi(\theta|y, \xi)$ (10,000, 30,000, 50,000 samples for the MINE estimator); $I_\epsilon$ estimators (right column): 100 $\theta$ samples from $p(\theta)$ and 10 y samples from each $\theta_i$ and 30 posterior samples for the marginal likelihood distribution (1000 $\theta$ and 10 y samples for the MINE estimator).

### 7.2.2 Preference

Another benchmark that we use to test the effectiveness of $I_\epsilon(p,\xi)$'s log-sum-exp stabilization is Preference test. The Preference experiment is designed to understand consumer behavior with a utility function [Samuelson, 1948, Foster et al., 2019]. The experiment provides the proposal to subjects and checks their preference as an output to use in the utility function.

We use the experiment from [Foster et al., 2019]. We start from different reference prior distribution using $-7.35$ as the mean instead of 0. The KL divergence from the original prior distribution is $\epsilon = 0.2$. VNMC estimator in our test shows more error with a small number of samples M = 30 (fig. 4, top left), while ACE estimator shows accurate estimation for the EIG with small number of marginalization samples (fig. 4, middle left). MINE estimator, on the other hand, estimate the EIG lower than the true EIG (fig. 4, bottom left). But the optimal experimental set-up are same with the true EIG.

We further estimate $\hat{I}_\epsilon^{\text{VNMC}}$ and $\hat{I}_{\epsilon,\max}^{\text{ACE}}$ estimator using only M = 30 samples and MINE, with an increasing series of ambiguity set radii $\epsilon$ (fig. 4, right). By applying ambiguity set 0.01, the $I$ estimation boundary becomes tight. If we apply ambiguity set $\epsilon = 0.1$, the lower bound (ACE) becomes higher than the true $I$ while the upper bound (VNMC) becomes lower than the True $I$ (fig. 4, right). The $\hat{I}_{\epsilon,\max}^{\text{ACE}}$ and $\hat{I}_\epsilon^{\text{VNMC}}$ are no longer the corresponding upper bound and lower bound. $\hat{I}_{\epsilon,\max}^{\text{MINE}}$ with $\epsilon = 0.1$ increase the EIG estimation and change the optimal experimental set up (fig. 4, bottom right).

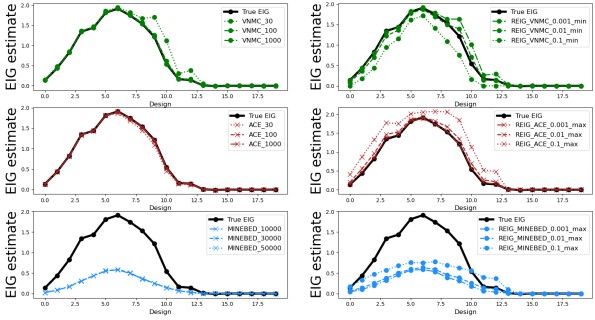

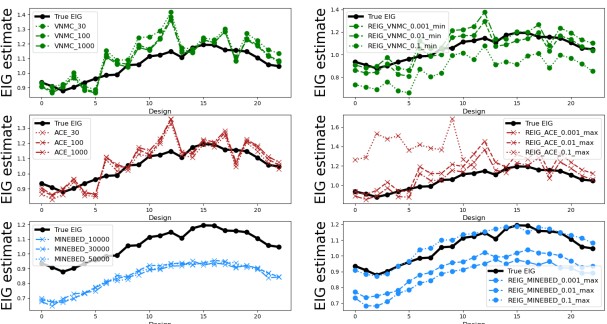

Figure 4: Preference test convergence for EIG estimators (left column): 100*10 samples from $p(\theta, y)$ and 30/100/1000 samples from $q_\phi(\theta|y,\xi)$ (10,000, 30,000, 50,000 samples for the MINE estimator); $I_\epsilon$ estimators (right column): 100 $\theta$ samples from $p(\theta)$ and 10 y samples from each $\theta_i$ and 30 posterior samples for the marginal likelihood distribution (1000 $\theta$ samples and 10 y samples for MINE estimator).

Figure 5: Pharmacokinetic model convergence for EIG estimators (left column): 100*10 samples from $p(\theta, y)$ and 30/100/1000 samples from $q_\phi(\theta|y,\xi)$ (10,000, 30,000, 50,000 samples for the MINE estimator); $I_\epsilon$ estimators (right column): 100 $\theta$ samples from $p(\theta)$ and 10 y samples from each $\theta_i$ and 30 posterior samples for the marginal likelihood distribution (1000 $\theta$ samples and 10 y samples for MINE estimator).

### 7.2.3 Pharmacokinetic model

The last benchmark that we use to test the effectiveness of $I_\epsilon(p, \xi)$ is Pharmacokinetic study. The Pharamcokinetic study is designed to understand how the medicine is absorbed, distributed and eliminated in the subject's body, which we can understand with the compartmental model. We can estimate the compartment model's parameter by blood sampling and we want to calculate the optimal blood sampling time. [Ryan et al., 2014, Kleinegesse and Gutmann, 2020]

We start from a different reference prior distribution with its mean taken to be $[0.1, \log 0.1 \log 20]$ whose KL divergence from the original prior distribution is 0.1. The accuracy of the estimators $\hat{I}^{\text{VNMC}}$ and $\hat{I}^{\text{ACE}}$ in our tests have increases with M (fig. 5, top left, middle left) while $\hat{I}^{\text{MINE}}$, on the other hand, does converge to a substantial underestimate (fig. 5, bottom left), yet identifies the same optimal experiment.

We further compute $\hat{I}_\epsilon^{\text{VNMC}}$ and $\hat{I}_{\epsilon,\max}^{\text{ACE}}$ estimators using only M = 30 samples and $\hat{I}_{\epsilon,\max}^{\text{MINE}}$, with an increasing series of ambiguity set radii $\epsilon$ (fig. 5, right). By applying ambiguity set 0.1, the lower bound (ACE) becomes higher than the true $I$ while the upper bound (VNMC) becomes lower than the True $I$ (fig. 5, right). The $\hat{I}_{\epsilon,\max}^{\text{ACE}}$ and $\hat{I}_\epsilon^{\text{VNMC}}$ are no longer the corresponding upper bound and lower bound. MINE estimator with ambiguity set $\epsilon = 0.1$, on the other hand, improve the EIG estimation so that the estimation is close to the true EIG (fig. 5, bottom right).

## 8 DISCUSSION

This work presents an introduction and initial numerical experiments testing the use of a robust modification of expected information gain as a criterion for Bayesian model selection. The $I_\epsilon$ estimator is designed to both have rigorously defined approximation properties (as the minimization of a tangent approximation to the EIG over a convex ambiguity set), and to yield a practical algorithm in practice (algorithm 1, which post-processes the samples of previously defined estimators by the solution of a 1D convex optimization problem).

While our initial results are promising, our understanding of how to apply this method is not complete at this time. Most of the remaining questions relate to the choice of the radius $\epsilon$ of the ambiguity set over which the approximated EIG is taken to be robust. A definite a priori estimate for $\epsilon$ seems unlikely in most cases.

In an approach analogous to Morozov's discrepancy principle, a logical choice of $\epsilon$ would be one such that the implied radius of the ambiguity set is on the same order as the error that the sampling based estimator (such as VNMC or ACE) has introduced into the problem. The fact that these two estimators provide (in expectation) an upper and lower bound for the true EIG of a design suggests that there may be a way to combine the two estimators into an a posteriori estimate of the proper choice of $\epsilon$. Further investigation into this topic is the subject of future research.

## Acknowledgements

We would like to thank Alexander Shapiro for several stimulating discussions on ambiguity set.

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

Ruoyong Yang and James O Berger. *A catalog of noninformative priors*.

# A   EXPERIMENTS DETAILS

## A.1   IMPLEMENTATION

All experiments were implemented in PyTorch 1.4.0 [Paszke et al., 2019] and Pyro 0.3.4 [Bingham et al., 2018].

## A.2   A/B TEST

The reference prior and likelihood for the A/B test were as follows:

$$\theta \sim \mathcal{N}(\begin{pmatrix} 0 \\ 0 \end{pmatrix}, \begin{pmatrix} 10^2 & 0 \\ 0 & 1.82^2 \end{pmatrix})), y|\theta, \xi \sim \mathcal{N}(X_\xi \theta, I) \quad (25)$$

## A.3   PREFERENCE

We use the utility function from [Foster et al., 2019].

$$\text{let } \xi \in \mathrm{R} \quad (26)$$
$$\theta \sim \mathcal{N}(-7.35, 20^2) \quad (27)$$
$$\eta|\theta, \xi \sim \mathcal{N}(\xi - \theta, 1 + |\xi|^2) \quad (28)$$
$$y = f(\eta) \quad (29)$$
$$f : \mathrm{R} \to [\epsilon, 1 - \epsilon] \quad (30)$$
$$x \to \begin{cases} \epsilon & \text{if } x \le \text{logit}(\epsilon) \\ 1 - \epsilon, & \text{if} x \ge \text{logit}(1 - \epsilon) \\ \frac{1}{1 - e^{-x}} & \text{otherwise,} \end{cases} \quad (31)$$

## A.4   PHARMACOKINETIC MODEL

The prior distribution and noise distributions for the pharmacokinetic model are defined as follows,

$$\begin{pmatrix} k_a \\ k_e \\ V \end{pmatrix} \sim \log \mathcal{N} \left[ \begin{pmatrix} 0.1 \\ \log 0.1 \\ \log 20 \end{pmatrix}, \begin{pmatrix} 0.05 & 0 & 0 \\ 0 & 0.05 & 0 \\ 0 & 0 & 0.05 \end{pmatrix} \right] \quad (32)$$

$$y_t = \frac{400}{V} \frac{k_a}{k_a - k_e} (\exp -k_e t - \exp -k_a t)(1 + \epsilon_{1t}) + \epsilon_{2t}, \quad (33)$$

where $\epsilon_{1t} \sim \mathcal{N}(0, 0.01), \epsilon_{2t} \sim \mathcal{N}(0, 0.1)$, and $k_a > k_e$.