# OpenReview forum: "Robust Expected Information Gain for Optimal Bayesian Experimental Design Using Ambiguity Sets"
_auai.org/UAI/2022/Conference — UAI 2022 Oral_

### Official Review · Reviewer_QHtK · 2022-04-07

**Q2(1) Originality/Novelty:** 3
**Q2(2) Significance/Impact:** 2
**Q2(3) Correctness/Technical Quality:** 3
**Q2(6) Clarity Of Writing:** 3
**Q6 Overall Score:** 7
**Q8 Confidence In Your Score:** 3

**Q1 Summary And Contributions:**

If the Bayesian scientist has a precise prior, an experimental design could be selected by choosing the design that maximizes expected information gain. However, the scientist may not have a precise prior. The paper proposes a minimax approach to Bayesian experimental design: pick the design that maximizes minimum expected information gain, where the minimum is taken over prior distributions that are "nearby" a reference prior. Technical proposals are made to implement this strategy.





**Q2 Assessment Of The Paper:**

More detailed information regarding each of these aspects is given below:

**Q2(4) Quality Of Experiments (Optional):**

3: Good: The experimental evaluation is adequate, and the results convincingly support the main claims.

**Q2(5) Reproducibility:**

2: Fair: Key resources (e.g., proofs, code, data) are unavailable but key details (e.g., proof sketches, experimental setup) are sufficiently well-described for an expert to confidently reproduce the main results.

**Q3 Main Strengths:**

- The submission addresses an interesting and important problem in a sophisticated way.
- Clever application of complex optimization renders this important problem solvable.
- Proposal is supported with promising experimental results.

**Q4 Main Weakness:**

- Some technical and conceptual matters could be sharpened.
- The virtues of the proposal over alternatives are not communicated in a clear way.


**Q5 Detailed Comments To The Authors:**

If the Bayesian scientist has a precise prior, an experimental design could be selected by choosing the design that maximizes expected information gain. However, the scientist may not have a precise prior. The paper proposes a minimax approach to Bayesian experimental design: pick the design that maximizes minimum expected information gain, where the minimum is taken over prior distributions that are "nearby" a reference prior. Technical proposals are made to implement this strategy. This is perhaps a strange mixture of Bayesian and non-Bayesian decision principles, but I imagine it is not without precedent in the literature on imprecise probability. The subject is interesting and worthy, although I am not completely able to evaluate the novelty or feasibility of the proposed approach. The authors should clarify the advantages of this approach over alternatives and sharpen some technical points.

To implement the idea we have to:

For every experimental design $\xi$, find the worst case expected information gain, over priors $q(\theta)$ nearby to a reference prior $p(\theta)$. The authors write that it is not plausible to do this by a nested Monte Carlo method (how much worse would the naive approach be?) and propose a solution via complex analysis. As far as I understand, the idea is the following: the ambiguity set $A(\epsilon, p)$ is a convex subset. So if EIG were an affine function, finding the worst-case distribution $q$ for a given design $\xi$ would be a tractable convex optimization problem. However, EIG is not an affine function (why? It is the expectation of a measurable function, is that not sufficient? If it isn’t why does the third paragraph in Section 4 suggest it is?) So instead, we find an affine approximation to EIG, $I_{aff}$, that is a good approximation inside of the ambiguity set by (13). Now we have a convex optimization problem. Then, invoking a result of Shapiro (2017), we convert this into a dual problem in a one dimension variable $\lambda$. (why is this a precise solution to the original problem and not an approximation? Or: how precise is this approximation?) So far so good: we have an efficient way of computing the worst case EIG over the ambiguity set for a given experiment.

Now we have to estimate the expectation term in the dual problem by a nested Monte Carlo method. The outer loop samples from the reference prior $p(\theta)$ and the inner loop computes, for a given $\theta$, the EIG of the design in question. There are three options for the inner loop: VNMC, ACE, MINE. The performance of these competitors is studied in a number of simulations, showing reasonably good performance at picking out the true minimax design.

I would like the following questions clarified, in order of importance.

1. Naively, one would approach this problem by adding another outer loop sampling uniformly over the ambiguity set. As I understand, the main innovation here is proposing a convex optimization approach that cuts out the need for this outer loop. Am I right about this? How much computation does this save? Does this make the difference between a feasible and infeasible solution to the problem? Has this been proposed elsewhere? I think this is the main selling point of the paper, so it should be made clearly and prominently.

2. In the third paragraph of Section 4, the author writes: “If the objective function of interest $f(q)$ is the expectation under $q(\theta)$ of a measurable quantity of interest $Z(\theta)$, then it is an affine function of $q$ and we may use duality to simplify the maximization or minimization of $E_{q(\theta)}[Z(\theta)]$ over $A(\epsilon, p(\theta))$ into a dual problem with only one variable $\lambda >= 0$.”

However, in section 5, the authors write: “we note that $I(q,\xi)$ is a quantity that is concave in $q$, … and the duality framework of section 4 cannot be applied directly.” This motivates the approximation of $I(q,\xi)$ by $I_{aff}$.

But what goes wrong? Isn’t $I(q,\xi)$ the expectation under $q$ of a measurable quantity and therefore affine according to the third paragraph of Section 4? Why the detour in Section 5.1? I am probably missing something obvious, but I would like the authors to spell it out for me.

3. Is the solution to the dual problem in (8) a *precise* solution to the problem in (7)? Or is it merely an approximation? I would like a clearer statement and appeal to the relevant result from Shapiro.

4. I would like a little bit more discussion of the background decision theory. Is there some precedent for this mixture of Bayesian and minimax, or worst-case, decision rules? (I imagine there is, in the imprecise probability literature. See, for a start, Section 3.2 here: https://plato.stanford.edu/entries/formal-belief/#ImpProb) Does someone argue for why this is a good decision procedure?

5. The authors are trying to solve a tricky problem, so it is not surprising that many approximations and computational kludges are necessary along the way. But can the authors suggest something about whether this procedure should pix out the true minimax solution? Are there reasons to think finite-sample guarantees can be given? Or can we expect only asymptotic results? I don't expect analytic results, but some discussion one way or the other.


Typos:

-Paragraph 2 of section 1: “information gain in computable in closed form” should be “information gain *is* computable …”

-Paragraph 1 of section 2.2: “marginial” should be “marginal”.


**Q7 Justification For Your Score:**

The submission strikes me as an innovative attempt on an important problem, with good experimental support.

**Q9 Complying With Reviewing Instructions:**

1: Yes.

---

### Official Review · Reviewer_oeoE · 2022-04-12

**Q2(1) Originality/Novelty:** 2
**Q2(2) Significance/Impact:** 2
**Q2(3) Correctness/Technical Quality:** 2
**Q2(6) Clarity Of Writing:** 2
**Q6 Overall Score:** 5
**Q8 Confidence In Your Score:** 3

**Q1 Summary And Contributions:**

The paper proposes a robust expected information gain (REIG), a modification of the objective in EIG maximization obtained by minimizing over a set of prior distribution which are within a certain radius of some reference distribution.

**Q2 Assessment Of The Paper:**

More detailed information regarding each of these aspects is given below:

**Q2(4) Quality Of Experiments (Optional):**

2: Fair: The experimental evaluation is weak: important baselines are missing, or the results do not adequately support the main claims.

**Q2(5) Reproducibility:**

2: Fair: Key resources (e.g., proofs, code, data) are unavailable but key details (e.g., proof sketches, experimental setup) are sufficiently well-described for an expert to confidently reproduce the main results.

**Q3 Main Strengths:**

The proposed REIG is defined as a maxmin optimization problem. The authors propose a way to approximate it.

**Q4 Main Weakness:**

The paper clarity should be improved for instance the following sentences are not clear
-"By the concavity of KL-divergence with respect to its first argument": are you referring to D_KL  in (10)?
-"Due to the data processing inequality", what is the data processing inequality?





**Q5 Detailed Comments To The Authors:**

The experimental setting in Section 7 is not very interesting for AI and machine learning. It would be more interesting to consider experiments where REIG is used for active learning (to learn an input-output function) and check how the robustness helps to improve the design. For instance, you could consider a set of Gaussian Process priors defined by a different mean function and define your KL distance on the mean function to get a class of Gaussian process priors. Then you could use REIG for active learning.

These types of KL-based robust models have previously been proposed and studied within the robust Bayesian community, see for instance "Approximate Models and Robust Decisions", James Watson and Chris Holmes, Section 2. These references should be added to the paper.

Minor:
-pag 2 "topology like TV" what is TV?

**Q7 Justification For Your Score:**

The proposed criterion may be useful for practical applications in AI and ML (active learning). The paper should show that the criterion is effective in this kind of applications of EIG.

**Q9 Complying With Reviewing Instructions:**

1: Yes.

---

### Official Review · Reviewer_PiER · 2022-04-13

**Q2(1) Originality/Novelty:** 4
**Q2(2) Significance/Impact:** 4
**Q2(3) Correctness/Technical Quality:** 3
**Q2(6) Clarity Of Writing:** 4
**Q6 Overall Score:** 9
**Q8 Confidence In Your Score:** 4

**Q1 Summary And Contributions:**

The authors address the problem of prior mis-specification in optimal Bayesian experimental idea.  The approach involves optimization over an "ambiguity set" (a ball of a certain radius in KL-divergence centered about the nominal prior), which is approximately solved via a relaxation and a reformulation of the optimization by passing to the dual.

**Q2 Assessment Of The Paper:**

More detailed information regarding each of these aspects is given below:

**Q2(4) Quality Of Experiments (Optional):**

3: Good: The experimental evaluation is adequate, and the results convincingly support the main claims.

**Q2(5) Reproducibility:**

3: Good: Key resources (e.g., proofs, code, data) are available and key details (e.g., proofs, experimental setup) are sufficiently well-described for competent researchers to confidently reproduce the main results.

**Q3 Main Strengths:**

The description of the "prior uncertainty" problem is laid out very clearly, and the proposed approach is very elegant -- both in the basic formulation and in the solution via relaxation plus passing to the dual.

**Q4 Main Weakness:**

As the authors note, they do not yet have much theory for how to choose the radius of the ambiguity set.

**Q5 Detailed Comments To The Authors:**

This was a very nice paper, and a pleasure to read.  Thank you.

**Q7 Justification For Your Score:**

This was by far my favorite of the papers that I read.  I think the key idea is important and beautifully addressed (and the issue of prior mis-specification is far too often overlooked in Bayesian methods).

**Q9 Complying With Reviewing Instructions:**

1: Yes.

---

### Decision · Program_Chairs · 2022-05-15

**Decision:**

Accept (Oral)

**Comment:**

Meta Review: The authors address the problem of prior misspecification in optimal Bayesian experimental design. If the Bayesian scientist has a precise prior, an experimental design could be selected by choosing the design that maximizes expected information gain. However, the scientist may not have a precise prior. The paper proposes a minimax approach to Bayesian experimental design: pick the design that maximizes minimum expected information gain, where the minimum is taken over prior distributions that are "nearby" a reference prior; the distance here is defined using KL-divergence. The minimax problem is solved approximately via relaxation and a reformulation of the optimization by passing to the dual.

All reviewers agree that the problem studied is interesting and important, and the approach proposed is very elegant. The proposal is also supported by promising experimental results. There is a mix of ratings, but all the reviews are positive.

An open problem left is how to choose the radius of the ambiguity set. The authors acknowledge that this is a limitation.

The potential weakness is that the virtues of the proposal over alternatives could be better communicated. There are also some minor concerns regarding the use of abbreviations that should not be assumed known.

Overall I feel this is a clear accept, with potential for oral presentation. Reviewer oeoE only suggests borderline accept, but the concerns raised there seem minor to me.